# Development of deep learning algorithms for predicting blastocyst formation and quality by time-lapse monitoring

Qiuyue Liao [1,3], Qi Zhang[2,3], Xue Feng[1,3], Haibo Huang[2,3], Haohao Xu[2,3], Baoyuan Tian[2], Jihao Liu[2], Qihui Yu[2], Na Guo[1], Qun Liu[1], Bo Huang[1], Ding Ma[1], Jihui Ai [1,4✉], Shugong Xu [2,4✉] & Kezhen Li [1,4✉]

Approaches to reliably predict the developmental potential of embryos and select suitable embryos for blastocyst culture are needed. The development of time-lapse monitoring (TLM) and artificial intelligence (AI) may help solve this problem. Here, we report deep learning models that can accurately predict blastocyst formation and usable blastocysts using TLM videos of the embryo's first three days. The DenseNet201 network, focal loss, long short-term memory (LSTM) network and gradient boosting classifier were mainly employed, and video preparation algorithms, spatial stream and temporal stream models were developed into ensemble prediction models called STEM and STEM$^+$. STEM exhibited 78.2% accuracy and 0.82 AUC in predicting blastocyst formation, and STEM$^+$ achieved 71.9% accuracy and 0.79 AUC in predicting usable blastocysts. We believe the models are beneficial for blastocyst formation prediction and embryo selection in clinical practice, and our modeling methods will provide valuable information for analyzing medical videos with continuous appearance variation.

[1] Department of Gynecology and Obstetrics, Tongji Hospital, Tongji Medical College, Huazhong University of Science and Technology, Wuhan, Hubei, China. [2] Shanghai Institute for Advanced Communication and Data Science, Shanghai University, Shanghai, China. [3]These authors contributed equally: Qiuyue Liao, Qi Zhang, Xue Feng, Haibo Huang, Haohao Xu. [4]These authors jointly supervised this work: Jihui Ai, Shugong Xu, Kezhen Li. ✉email: jihuiai@tjh.tjmu.edu.cn; shugong@shu.edu.cn; tjkeke@126.com

mproving the efficacy of in vitro fertilization (IVF) has always been a focused issue, in which embryo selection and transfer are key procedures. Clinically, embryos are evaluated by embryologists and transferred at cleavage stage on day 3 (D3) or blastocyst stage on day 5 or 6 (D5/6) post-fertilization. Blastocyst transfer allows self-selection of embryos with higher developmental potential, thus can maximize the implantation rates and live birth rate in fresh cycles[1,2]. However, it may lead to cycle cancellation due to failure of embryo development to blastocyst stage, causing mental stress to both patients and embryologists. Therefore, selecting embryos suitable for extended culture can help embryologists improve implantation rates, decrease transfer cancellation rate, and reduce mental stress. In IVF clinics, morphological features and development rate are key indicators for embryo selection. Traditionally, embryos are moved out of the incubators at several discontinuous time points and empirically evaluated by embryologists[3,4], which is subjective and unreviewable. More importantly, it may overlook some vital biological information during the dynamic embryo development event. Hence, traditional morphology assessment achieves a relatively low IVF success rate, and the clinical pregnancy rate per transfer is approximately 35%[5]. Time-lapse monitoring (TLM) is an emerging, powerful tool for embryo assessment and selection, where the embryos are cultured in incubators with built-in microscopes to automatically obtain images every 5–20 min at a certain focus and magnification. Thus, this technology can continuously monitor the dynamic development event without disturbing the culture environment and provide reviewable and stable video data for embryo selection[6,7].

Although TLM may improve the success rate of embryo selection, the surge of embryo data represents significant challenges in vision-based analysis. The FDA-approved software Eeva[TM] (Early Embryo Viability Assessment) is an embryo selection algorithm (ESA) based on TLM data, which provides blastocyst formation prediction by measuring cell division timings P2 (time between cytokinesis 1 and 2) and P3 (time between cytokinesis 2 and 3)[6]. The adjunctive use of traditional D3 morphology evaluation plus Eeva[TM] showed an evident improvement in embryo selection compared with D3 morphology evaluation alone[8–10]. However, compared to manual annotation of the same embryo videos, Eeva[TM] showed no superiority for blastocyst prediction and embryo selection[11]. Various ESAs have been published to link different parameters with blastocyst formation, implantation, or pregnancy. However, Barrie et al. applied six published ESAs to a large set of known implantation embryos, and none of these ESAs surpassed an AUC (area under the receiver operating characteristic (ROC) curve) of 0.65 (0.54–0.63), indicating poor diagnostic value[12]. Therefore, there is a strong demand for a method to improve embryo selection using the copious amount of available TLM data.

Recently, artificial intelligence (AI) techniques, especially deep learning models, have made significant advance in big data feature learning. Human expert-level or even better achievements of deep learning have been reported in the screening and diagnosis of diseases with medical images[13–16]. To date, several studies have employed deep learning algorithms for embryo quality grading or development stage classification based on static images from TLM[17–21]. However, few studies have explored deep learning methods for directly analyzing TLM videos. To the best of our knowledge, one deep learning model (IVY) studied whole embryo videos to provide predictions of pregnancy[22], but the modeling methods were vaguely described. Although video analysis has been a highly active topic in AI, the study of embryo videos still faces significant difficulties due to the appearance variation and occlusion in cell division, which changes continuously and is difficult to track. Various attempts are required to efficiently incorporate deep learning algorithms and TLM videos to provide reliable methods for embryologists to select good-quality embryos, and thus to help improve the success rate of IVF.

Based on these insights, we aim to establish a model based on TLM videos and deep learning algorithms for accurately predicting blastocyst formation and blastocyst quality on D3. We believe this model will provide valuable information for clinical decisions about blastocyst culture and embryo selection.

## Results

**Datasets and procedures**. The procedure of video selection is shown in Fig. 1a. A total of 26,113 embryos from 2594 IVF and intracytoplasmic sperm injection (ICSI) cycles were cultured in TLM incubators from February 2014 to December 2017, and the women's age at the retrieval cycles ranged from 20 to 50 years old (30.56 ± 5.03 years). After the D3 transferred/cryopreserved/discarded embryos elimination and the initial quality review, 12,912 videos were retained. Then, 1319 videos were randomly selected and labeled for video preparation algorithms. The remaining 11,593 videos were screened by video length (frames longer than 750 were retained) and pronuclei fading (PNF) recognition (e.g., indecipherable or obscured were eliminated). Ultimately 10,432 videos were retained for further analysis.

To develop the prediction models, as shown in Fig. 1b, 577 videos were randomly chosen from the 1319 embryos to build the cell-counting mode. Videos for prediction model analysis were divided into training ($n = 8346$) and validation datasets ($n = 2086$) to build the spatial stream model and the temporal stream model. Finally, the spatial–temporal ensemble model (STEM) was developed by weight average and then compared with embryologists. To improve the clinical application of our algorithm, a model STEM[+] that predicted usable blastocyst formation was further developed based on the spatial–temporal ensemble procedures. The clinical pregnancy outcomes of predicted embryos were calculated to evaluate the efficacy of our model.

**Video preparation algorithms and PNF recognition**. Both standard IVF- and ICSI-fertilized embryos were included in our study. Previous studies have shown a statistically significant difference in cleavage time between IVF and ICSI embryos, and the use of PNF rather than insemination as the starting time point can minimize the variations in recording timings, thus enabling simultaneous analysis of the IVF and ICSI embryos[23–25]. Therefore, 1-cell recognition, pronuclei (PN) estimation, and PNF recognition algorithms were developed to unify the start time of IVF and ICSI embryo videos.

Figure 2a shows the male PN and a female PN before the first cleavage. The 1-cell recognition algorithm to distinguish 1-cell stage with multicell stage achieved an accuracy of 99.4% in the 264 videos (Supplementary Table 1). Then, the accuracy of PN estimation algorithm to recognize PN in 1-cell stage was 92.9% (Fig. 2b). After removing the noisy segment by a correction mechanism, the PN estimation model achieved an accuracy of 93.4% with 94.8% sensitivity and 90.9% specificity (Supplementary Table 2).

PNF, which is the fading of PN, occurs when the male and female PN fuse (Fig. 2a). Of the 264 embryos, 126 were IVF-fertilized, and the PNF mainly ranged from 150 to 300 frames (mean: 235), while 138 were ICSI-fertilized, and most PNF frames ranged from 200 to 350 frames (mean: 271) (Fig. 2c). The PNF recognition algorithm obtained an accuracy of 97.7% in these embryos (Fig. 2d).

**Cell-counting algorithm and cell number identification**. Morphokinetic parameters, such as duration of cytokinesis and

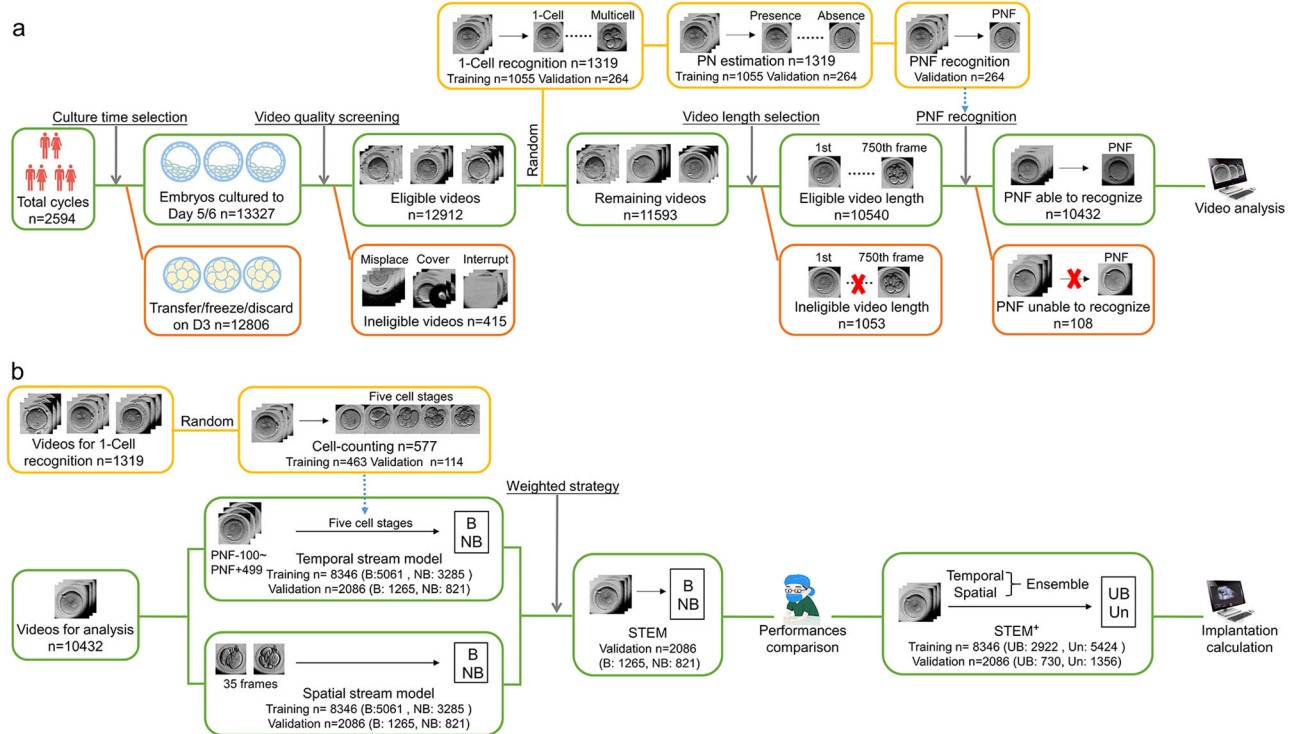

**Fig. 1 The workflow of video selection and video utilization. a** The workflow of video preparation for the prediction models. **b** The workflow of video analysis in the prediction models. Green boxes represent the reserved videos after each procession, red represents the discarded videos, and yellow represents the randomly separated videos. B blastocyst, NB nonblastocyst, UB usable blastocyst, Un unusable embryos.

temporality of various cell stages, are closely related to the development or pregnancy potential of embryos[26]; however, it is difficult for AI to automatically measure the morphokinetic parameters due to appearance variation in cell division (e.g., number of cells increases, cell shape deforms). To solve this problem, we established a cell-counting model to enable the automated classification of cell stages; hence, the morphokinetic parameters could be obtained by analyzing the changes of cell stages in consecutive frames.

As shown in Fig. 3a, the cell-counting algorithm learned the frames of five cell stages in 463 embryos, including 1-cell stage, 2-cell stage, 3-cell stage, 4-cell stage, and ≥5-cell stage (Fig. 3a). Then, this model was validated in an independent dataset comprised of 114 labeled videos. A total of 80,300 frames were obtained, and the percentage of frames were 42%, 16%, 6%, 18%, and 18% for the 1-cell, 2-cell, 3-cell, 4-cell, and ≥5-cell stages, respectively (Fig. 3b). The cell-counting model achieved an overall accuracy (fraction of correct frames) of 94.6% with sensitivity values of 97.2%, 88.3%, 89.7%, 92.8%, and 97.5% for 1-cell, 2-cell, 3-cell, 4-cell, and ≥5-cell stages, respectively (Fig. 3c and Supplementary Table 3).

**Temporal stream model and performance verification.** The established cell-counting algorithm was combined with the the long short-term memory (LSTM) network to develop a temporal stream model, which converted cleavage information into numerical information and then learned the duration and dynamic of cell numbers, thus predicting blastocyst formation based on morphokinetic parameters. In the videos for prediction model analysis ($n = 10432$), 600 frames ranging from PNF-100 to PNF+499 were extracted from each video to train the temporal stream model, and the accuracy of blastocyst formation prediction was 76.9%, and the AUC was 0.77 (Fig. 4a and Supplementary Table 4).

**Spatial stream model and performance verification.** Morphological assessment has long been the primary method used to distinguish embryo quality and development potential. In clinics, several timings of observation are required for the morphological assessment, including fertilization check, syngamy check, early cleavage check, day-2, day-3, day-4, and day-5 assessments[3]. We choose the five observation points in the first 3 days of embryo development and extracted seven frames at each point for modeling, including from PNF-75 to PNF-69 (fertilization check), PNF±3 (syngamy check), from PNF+33 to PNF+39 (early cleavage check), from PNF+249 to PNF+255 (D2 assessment), and from PNF+493 to PNF+499 (D3 assessment). After training the dataset ($n = 8346$) allocated from videos for prediction model analysis, we obtained 70.0% accuracy and 0.76 AUC for the prediction of blastocyst formation in the validation dataset ($n = 2086$) (Fig. 4a and Supplementary Table 4).

**Ensemble model and performance verification.** It is important to take into account that morphokinetic parameters and morphological features are two complementary markers for embryo selection and blastocyst prediction, so we integrated the spatial and temporal information to accurately predict blastocyst formation. A weighted average was used by traversing the weight between 0 and 1 at 0.01 intervals, and the optimal result was obtained by giving the weight of 0.66 in the temporal model and 0.34 in the spatial model (Fig. 4b). The final ensemble model STEM achieved 78.2% accuracy for prediction of blastocyst formation in the validation dataset ($n = 2086$) with 85.9% sensitivity and 66.3% specificity (Supplementary Table 4). The AUC of our STEM model was 0.82 (Fig. 4a).

**Embryologist prediction process and performance comparison.** To examine the performance of STEM, four embryologists were separately asked to provide blastocyst formation predictions in

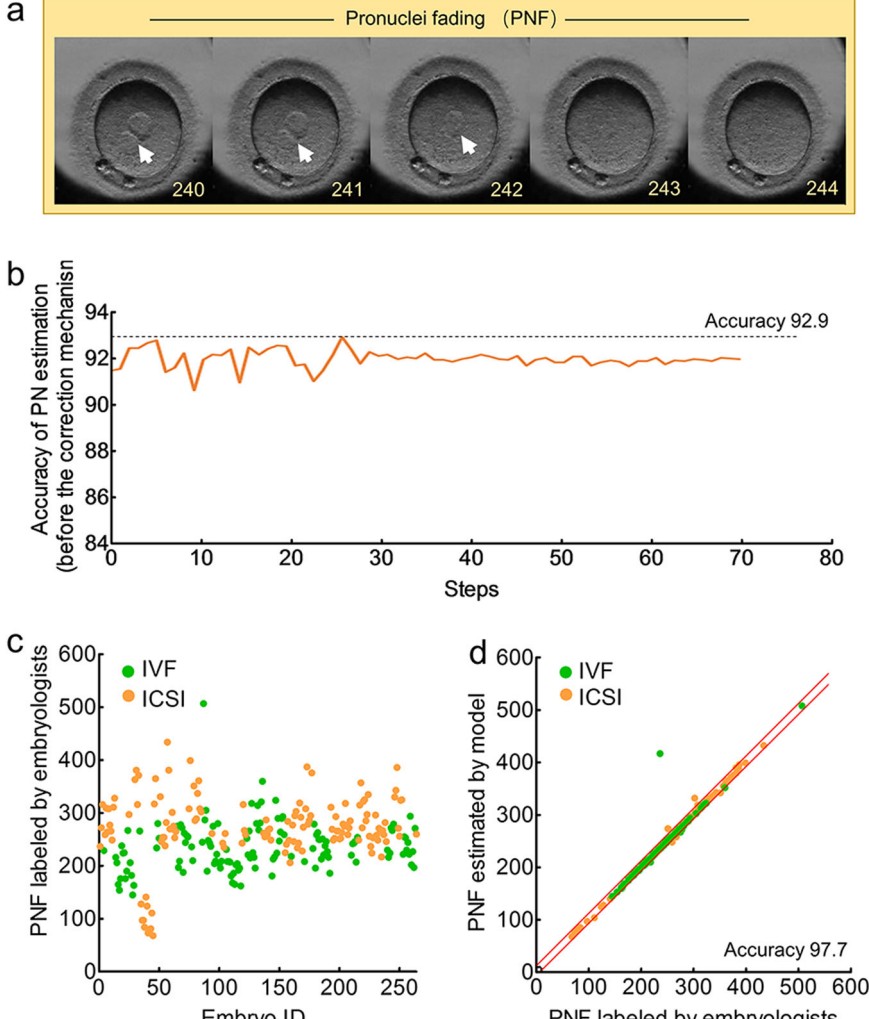

**Fig. 2 Performance of the PN estimation algorithm and PNF recognition in 264 videos. a** An example of PN and PNF in an embryo video. The arrows indicate the PN. The numbers in the right corner in each frame represent the frame number (start from the recording time). **b** Accuracy of the PN estimation model before the correction mechanism using a filtering algorithm. Accuracy is plotted against the training step during the length of 70 training steps. **c** The frames of PNF labeled by embryologists. **d** The correlation of frames between PNF labeled by embryologists and PNF estimated by AI. The red lines represent the boundary of 10 deviation PNF frames between AI and embryologists.

the validation dataset ($n = 2086$). The accuracies of the four embryologists were 67.8%, 64.5%, 65.5%, and 64.9%, separately (Supplementary Table 5).

The performances of embryologists were compared with STEM. Figure 4c shows the prediction outcomes of 2086 videos from the model and embryologists. The sensitivity/1-specificity points of embryologists trended below the ROC of STEM (Fig. 4d).

**Usable blastocyst prediction and performance evaluation.** Clinically, embryos cultured to D5/6 are graded by embryologists, and usable blastocysts are chosen to transfer or vitrification. To help embryologists select embryos that can be transferred or vitrified on D5/6, we further attempted to predict usable blastocyst formation using our modeling procedures. Videos for prediction model analysis ($n = 10,432$) were labeled anew as usable or unusable by three experienced embryologists, then 8346 (usable: 2922, unusable: 5424) were allocated into the training dataset and 2086 (usable: 730, unusable: 1356) into the validation dataset. In the training dataset, temporal stream and spatial stream information was separately learned as mentioned above, and the accuracies in validation dataset achieved 71.8% in

temporal and 68.6% in spatial model. Ensemble model STEM[+] was obtained by giving the weight of 0.63 in the temporal model and 0.37 in the spatial model, which reached 71.9% accuracy and 0.79 AUC in predicting usable and unusable blastocysts formation in validation dataset (Fig. 5a and Supplementary Table 6).

As usable blastocysts can be transferred to achieve pregnancy, we followed the clinical pregnancy outcomes of the transferred blastocysts in the validation dataset. Among the 730 usable blastocysts, only 209 blastocysts had known implantation outcomes (number of gestational sacs matched the number of transferred embryos). A total of 160 embryos were predicted as usable blastocysts by STEM[+], of which 81 implanted and 79 failed. Another 49 embryos were predicted as unusable blastocysts by STEM[+], and among them 21 implanted and 28 failed (Fig. 5b). The evidently higher implantation rate in predicted usable group (50.6% vs. 42.9%) indicated that STEM[+] may help embryologists select embryos with higher development potential.

## Discussion
In this study, we report deep learning models that can predict blastocyst formation (STEM) and usable blastocyst (STEM[+])

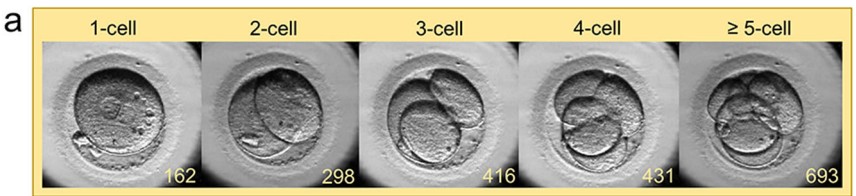

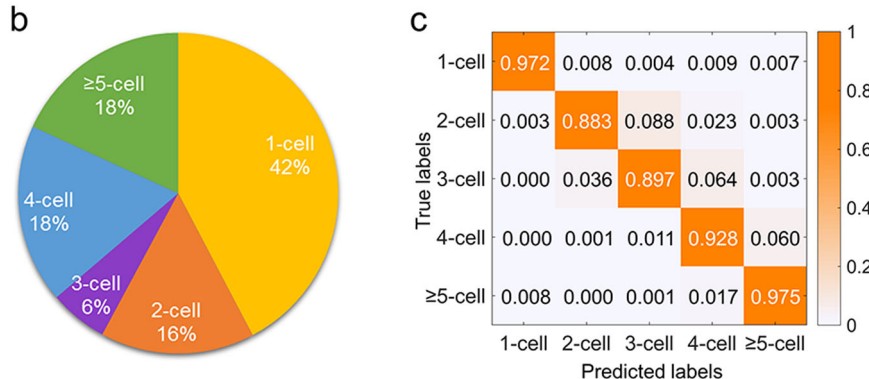

**Fig. 3 Performance of the cell-counting algorithm in 114 videos. a** An example of five cell stages in an embryo video. The number in the right corner in each frame represents the frame number (start from the recording time). **b** The proportions of frames in different cell stages labeled by embryologists. **c** The confusion matrix of the cell-counting algorithm for classifying frames into different cell stages. The matrices contain values of the model-labeled frames against true labeled frames and are colored based on the relative value for each cell stage.

using videos from the first 3 days of an embryo acquired by TLM. The AUCs are 0.82 of STEM and 0.79 of STEM$^+$, which are likely to be superior to previous ESAs. To the best of our knowledge, this study takes the lead in addressing medical videos with continuous appearance variation using deep learning algorithms.

To date, several studies have used AI methods for embryo assessment or blastocyst grading. The published AI models that aimed to offer a prediction of embryo outcome utilized either TLM images[17,21] or videos[22] of D5/6 blastocysts to predict pregnancy or live birth, so the course and endpoints are different with our study. In consideration of the fact that a successful pregnancy or live birth is a combined consequence of embryo potential, transfer time, and maternal conditions[27], our study chose blastocyst as an endpoint, which not only aims to predict the developmental potential of embryos but also reduce confounding factors from maternal conditions and other external factors[28]. Hence, by utilizing a large amount of TLM videos, an objective and automatic approach was developed for predicting the developmental potential of embryos, which may represent a great breakthrough in employing deep learning in TLM video assessment.

To develop such an approach, first, we built 1-cell recognition and PN estimation algorithms using DenseNet201 network to recognize PN and PNF. DenseNet network is an excellent architecture among different convolutional neural networks (CNNs), offering several compelling advantages, including strengthened feature propagation, encouraged feature reuse, and reduced parameter number[29]. Several studies that employed the DenseNet network to recognize medical images have obtained state-of-the-art achievements[30–32]. In our study, algorithms based on the DenseNet201 network also achieved high performance. After implementing these algorithms for video preparation, videos that were longer than 750 frames and contained recognizable PNF were retained for prediction model analysis. The same amount of frames were extracted based on PNF from each video, which not only eliminated the cleavage time difference

between IVF and ICSI embryos but also provided unified feature length for the LSTM network and spatial network.

Using the cell-counting algorithm, we classified the video frames into five stages according to the cleavage characteristics, including 1-cell, 2-cell, 3-cell, 4-cell, and ≥5-cell stages, and these time points and durations are statistically associated with embryo quality and pregnancy rate[26]. Focal loss was applied to the foundation of the Densenet201 network to build this algorithm, which can focus learning on hard examples and reduce the weight of easy negatives in dealing with class imblance[33]. When compared with other published human embryonic cell-counting frameworks based on CNN, our algorithm showed remarkably increased efficacy in recognizing 3-cell and 4-cell stages but lower performance for the 2-cell stage. We found that 8.8% of 2-cell frames were recognized as 3-cell frames by our model. This finding may be because the proportion of 3-cell stage frames (only 6%) is much lower than other cell stages. This low number was inadequate for AI to learn its characteristics; hence, the system mistakenly classified the 2-cell stage as the 3-cell stage. Nevertheless, the approach achieved an overall accuracy of 94.6%, which is comparable to the results of other published frameworks[18,34,35].

On the basis of the cell-counting algorithm, the LSTM network was used to develop the temporal stream model. The LSTM network is a special type of recurrent neural network (RNN) that is capable of learning the forward and backward dependencies among the frames in time series data[36]. In the field of medical science, this network has demonstrated its superiority in dealing with temporal information. Here, we employed the LSTM to learn the cell stage information output from cell-counting model in consecutive frames, and the morphokinetic parameters of embryo development were subsequently obtained. The ensemble of the cell-counting algorithm and LSTM network was a bright spot in our modeling process, yielding significant improvements than exclusively using LSTM to directly learn the features from the CNN network (our own experimental data in STEM: accuracy

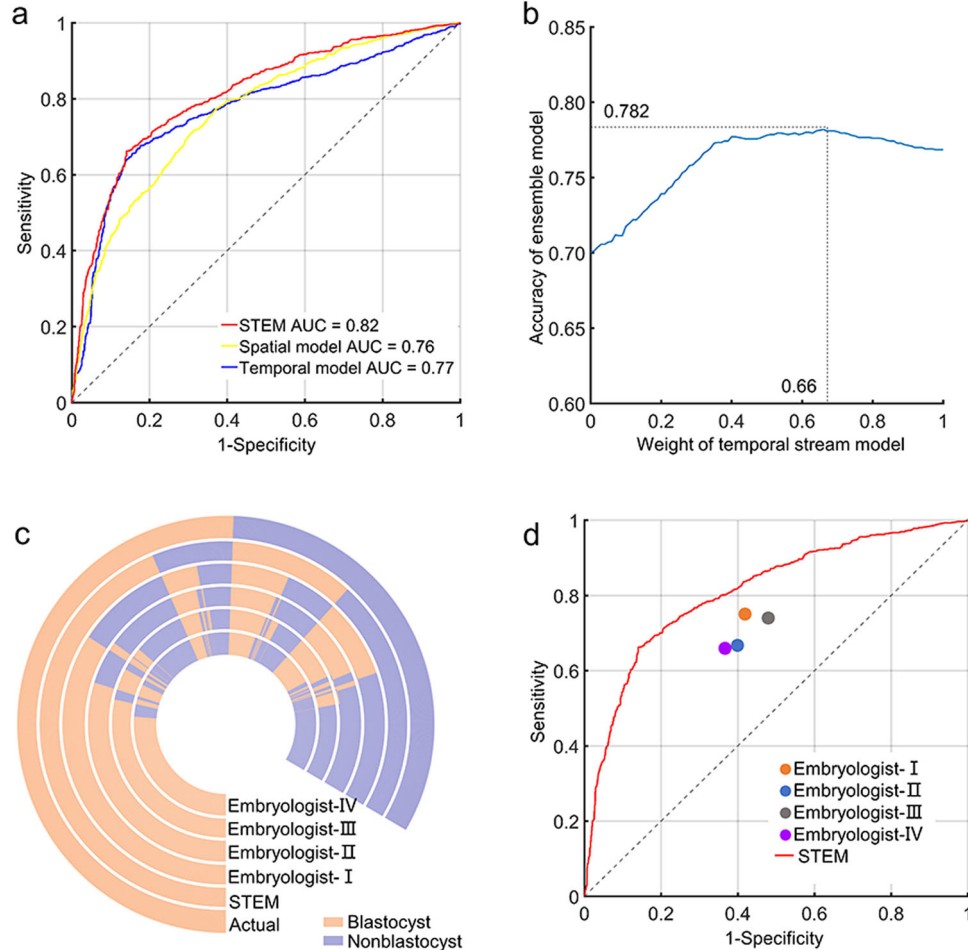

**Fig. 4 Performance of STEM in 2086 videos. a** Receiver operator characteristic (ROC) curves and area under the receiver operating characteristic curve (AUC) for spatial stream model (AUC = 0.76), temporal stream model (AUC = 0.77), and ensemble model STEM (AUC = 0.82). **b** Weighted average for combining the spatial stream and temporal stream models. The weight was obtained when the ensemble model achieved the highest accuracy. **c** The prediction outcomes of 2086 embryos obtained from STEM and embryologists. **d** The model's ROC curve and the sensitivity/1-specificity points of the embryologists.

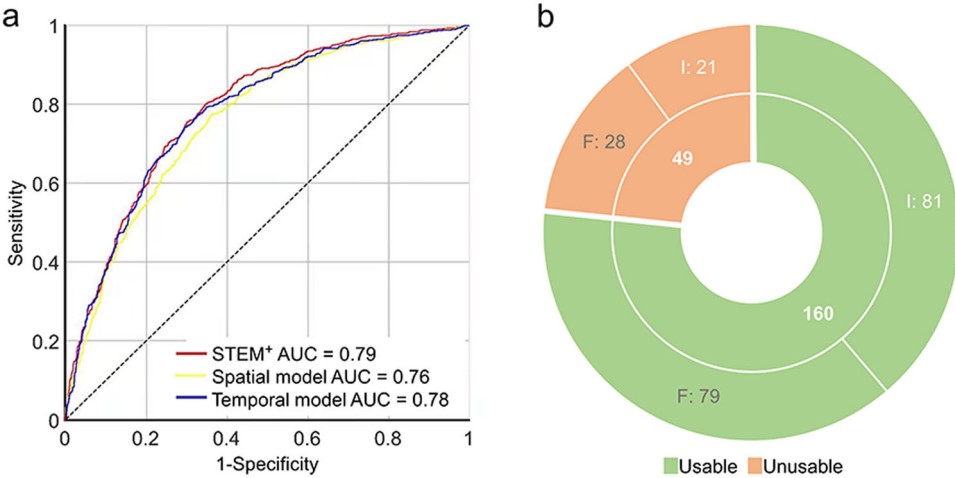

**Fig. 5 Performance of STEM+ in 2086 videos. a** Receiver operator characteristic (ROC) curves and area under the receiver operating characteristic curve (AUC) for spatial stream model (AUC = 0.76), temporal stream model (AUC = 0.78), and ensemble model STEM+ (AUC = 0.79). **b** The implantation outcomes of the transferred blastocysts in the validation dataset. A total of 160 embryos were predicted as usable blastocysts by STEM+ (green), and another 49 embryos were predicted as unusable blastocysts by STEM+ (orange). I implanted, F failed.

76.9% vs. 61%, sensitivity 84.7% vs. 85%, specificity 64.7% vs. 24%).

The spatial stream model based on the gradient boosting classifier was developed to learn the morphological features of the embryos. The gradient boosting classifier is a machine learning algorithm that has shown highly predictive performance in a wide range of practical applications and can identify the shortcomings of weak learners to optimize the model[37]. Studies have indicated that the gradient boosting classifier performs best among several machine learning algorithms in diagnosing epithelial ovarian cancer based on blood biomarkers[38], characterizing the risk of type 2 diabetes mellitus[39] and predicting vestibular dysfunction[40]. In our modeling process, the gradient boosting classifier also achieved higher performance than other tested machine learning algorithms with an accuracy of 70.0% in STEM.

The two-stream spatial–temporal network is a newly developed framework for video recognition in the field of deep learning that can capture both the object appearances from each frame and their motions along multiple frames in a video, thus seamlessly integrating object appearances with their motions for action recognition[41]. Methods that made use of the two-stream spatial–temporal network demonstrated state-of-the-art performances in video tasks, such as emotion recognition and action recognition[41–44]. In our study, the two-stream spatial–temporal ensemble predictive models STEM and STEM$^+$ yielded higher predictive efficacy than other published TLM-based ESAs. Eeva$^{TM}$ test, an FDA-approved algorithm, achieved an AUC of 0.728 for evaluating blastocyst formation as reported by Aparicio-Ruiz et al.[10], and a sensitivity of 58.8% and specificity of 84.2% in predicting usable blastocysts formation as reported by Conaghan et al.[9]. Cetinkaya et al. found that the cleavage synchronicity from 2 to 8 cells was the best predictor for blastocyst formation and quality with an AUC of 0.786, sensitivity of 83.43%, and specificity of 62.46%[45]. A prediction model based on the time to 2-cell stage, 3-cell stage, and 5-cell stage proposed by Milewski et al. obtained an AUC of 0.813 (sensitivity 75.4%, specificity 74.1%) in 271 embryos[46]. Our models were validated in a relatively large dataset ($n = 2086$) and showed higher superiority and reliability.

Compared with human embryologists, our models also achieved evidently higher performance either in predicting blastocyst formation or selecting high development potential blastocyst. STEM provided information on the feasibility of extended culture, thus helping embryologists to make evidence-based decisions and improve the success of IVF. STEM$^+$ predicted the usability of blastocysts, which added references for embryologists in selecting embryos with high development and implantation potentials. It is important to note that our models still have some limitations. First, they were trained on data obtained from single type of TLM in a single center, and the applicability to other TLMs and other centers remains unknown. However, our study employs a relatively large scale of embryos that have heterogeneous patient characteristics, which may compensate the single data source to some extent. But it is still necessary to execute multicentric studies to explore the universality. Second, we did not take clinical characteristics (e.g., parents' age and infertility reason) into account, so the influences of these factors on embryo development were ignored. A more accurate prediction model that incorporates clinical characteristics with video analysis should be established. Third, three-dimensional embryos were captured as two-dimensional images by TLM with only one focal depth, which made it difficult for AI to track the distorted or overlapped cells and then make a correct recognition or classification.

Taken together, our study develops a deep learning approach that can automatically and accurately predict blastocyst formation and quality based on videos of the first 3 days of an embryo using TLM. By developing the prediction models, we move a huge step forward toward making use of deep learning for analyzing medical videos with continuous appearance variation, providing valuable information to facilitate further studies on similar conditions. We believe this model is quite beneficial in clinics. Further application of this model can help select suitable embryos for extended culture and high potential blastocysts for transfer, while improving the efficacy of IVF. Further prospective studies are needed to extend this approach to clinics.

## Methods

**Data source**. This retrospective cohort study collected TLM videos and outcomes of embryos that were fertilized and cultured at Tongji hospital from February 2014 to December 2017. All couples who had embryos placed in TLM incubators (Primo Vision; Vitrolife) and further cultured to D5/6 were included regardless of the parents' age, infertility reason, ovarian stimulation protocol, and fertilization procedure.

In standard IVF cycles, fertilization was assessed 16–18 h after insemination, and normally fertilized oocytes with two PN were placed into the TLM. In the ICSI groups, the oocytes were placed in the TLM immediately after ICSI. The TLM captured images of embryos every 5 min until D3, after which approximately 750–800 images were obtained from each embryo and composed a video containing 750–800 frames. On D3, all embryos were moved out of the TLM incubators. Some embryos were chosen for transfer or cryopreservation. Extremely poor-quality embryos were discarded, while the remaining embryos were selected for culture until D5/6 to obtain blastocysts. This study recruited all the embryos cultured to D5/6, and these embryo videos were downloaded from the TLM.

This study was approved by the Ethical Committee of Tongji Hospital, Tongji Medical College, Huazhong University of Science and Technology (TJ-IRB20190317). In this retrospective study, we only obtained videos of embryos from clinical center, there were no situations of donation. And there was no identifiable information of patients. Informed consent was not required by the ethical committee.

**Video preparation**. To develop our model, we used python as program language. The frameworks were Keras and Scikit-learn, and the optimizer was Adam.

To unify the start time and video length for modeling, as shown in Supplementary Fig. 1a, we developed a 1-cell classification algorithm, a PN estimation algorithm, and a PNF recognition algorithm to preprocess the videos. In total, 1319 embryo videos were randomly selected for the video preparation process. Three embryologists with at least 5 years' experience evaluated the videos and labeled the PN and cleavage stages of embryo development in the videos (the label was assigned until at least two embryologists reached a consensus). In total, 80% of these videos were allocated to the training dataset, and 20% were allocated to the validation dataset.

*1-Cell recognition algorithm*. DenseNet201 network[29] was used to recognize the first cleavage in training data, and every frame was classified as 1-cell stage or multicell stage (Supplementary Fig. 1a). The efficacy of recognizing the 1-cell stage was tested in validation dataset.

*PN estimation algorithm*. DenseNet201 trained the frames in the 1-cell stage to learn the presence or absence of PN (Supplementary Fig. 1b). Then, a sequence containing 1 and 0 were generated from each video, where 1 represented the presence of PN and 0 the absence of PN. To improve the accuracy, the segment whose value was different from its neighborhood segments with consecutive same value was defined as a noisy segment, and a correction mechanism via a filtering algorithm was used to suppress the noisy segment in the output sequences.

---

For i = 1 : k
 Count the length of the noisy segment as num
 IF num==i
 Count the length of the segments with a consecutive same value at both ends of this noisy segment as j_left and j_right
 IF (j_left >= i and j_right>= i+1) or (j_right >=i and j_left >= i+1)
 Correct the value of noisy segment to the corresponding value of its left and right segments

Here: k = 6, and the result after the current (i-th) correction is the starting data for the next (i+1-th) correction.

---

This process was repeated until all the detectable disordered sequences were removed. After adjustment, the efficacy of the PN estimation model was measured in the validation dataset.

*PNF recognition algorithm*. The frame of PNF was computed by locating the last 1 in each output sequence from the PN model (Supplementary Fig. 1b). As PN fades

gradually, it was considered as a correct recognition if the deviation between the estimated value and labeled value was not greater than 10 frames.

**Video analysis**. We developed a model STEM to predict blastocyst formation and a model STEM$^+$ to predict usable blastocyst, the modeling datasets and procedures were the same between these two models. In total, 10,432 videos were labeled with outcomes by three embryologists with at least 5 years of experience, then they were divided into the training dataset (80%) and validation dataset (20%). Temporal and stream information of embryo videos was separately learned then combined as an ensemble prediction model. For STEM, the label of a video was blastocyst or nonblastocyst based on the Gardner's scoring system[47]. Specifically, embryos with blastocoel cavity formation were considered as blastocysts; otherwise, embryos were regarded as nonblastocysts (did not develop into blastocysts). For STEM$^+$, the label of a video was usable blastocyst or unusable blastocyst. Usable blastocyst was defined as the degree of expansion grade ≥3 and inner cell mass (ICM) and trophectoderm (TE) grading ≥BC or ≥CB according to the Gardner's scoring system. Embryos that did not develop into blastocysts or poor-quality blastocysts were defined as unusable blastocysts.

*Cell-counting model*. First, a cell-counting model was built to provide cleavage information for the temporal stream model. In total, 577 videos (randomly chosen from the 1319 embryos for video preparation) were divided into the training dataset (80%) and validation dataset (20%). Then, the DenseNet201 network and focal loss[33] were combined to learn the 1-cell, 2-cell, 3-cell, 4-cell, and ≥5-cell stages in the training dataset (Supplementary Fig. 1c), and every frame was output as a tag value that was composed of frame number and cell stage. The efficacy of this model was tested in the validation dataset.

*Temporal stream model*. A temporal stream model was built to learn the cleavage timing information of embryo development. A total of 10,432 embryo videos were used. Training data were input into the cell-counting model, and tag values composed of frame number and cell stage were obtained. Then, the LSTM network[36] was employed to learn both the tag values in temporal sequence and the relevant outcomes of embryos on D5 (Supplementary Fig. 1d). A temporal stream network was thus developed, and the prediction performance was obtained using validation dataset.

*Spatial stream model*. A spatial stream model was developed to capture the morphological features of the embryos. The training and validation datasets were the same as the temporal stream network. In total, 35 frames were extracted from each video based on the timings of embryo observations in the clinic, i.e., from PNF-75 to PNF-69 (fertilization check), PNF±3 (syngamy check), from PNF+33 to PNF +39 (early cleavage check), from PNF+249 to PNF+255 (D2 assessment), and from PNF+493 to PNF+499 (D3 assessment)[3]. Only 35 frames were chosen to avoid model overfitting. The DenseNet201 network extracted 1000-dimension spatial features of each frame under the order of observation timings. By orderly entering the $35 \times 1000$ features and the corresponding embryo's outcome on D5 into the gradient boosting classifier (Supplementary Fig. 1e)[37], a spatial stream model was established, and the accuracy for predicting blastocyst formation was analyzed by the validation dataset.

*Ensemble model*. Finally, an ensemble model was developed by integrating spatial and temporal networks to capture both morphology and cleavage information[48]. A weighted average was used to obtain an ensemble prediction value in 2086 videos (the validation dataset of spatial and temporal stream models) by multiplying the output prediction value of the spatial and temporal stream models by their weight and then adding these values. We traversed the weight of the temporal stream model between 0 and 1 at 0.01 intervals and defined the optimal weights when the accuracy of the ensemble prediction value reached the top. The performance of the ensemble model was evaluated in the 2086 videos (Supplementary Fig. 1f).

**Statistics and reproducibility**. To evaluate the performance of our algorithms, sensitivity (true-positive rate), specificity (true-negative rate), PPV (positive predictive value), NPV (negative predictive value), and accuracy values were measured in the validation datasets.

Sensitivity = True positive/Real positive
Specificity = True negative/Real negative
PPV = True positive/Predictive positive
NPV = True negative/Predictive negative
Accuracy = (True positive + True negative)/Total

ROC curves were generated by plotting the sensitivity against 1-specificity at various threshold settings. AUC was defined by calculating the area under the ROC curve.

**Embryologist validation**. Four experienced embryologists (engaged in this field for more than 10 years) from our center were asked to predict blastocyst formation for the validation dataset containing 2086 videos. Every embryologist reviewed the videos and provided an outcome for every embryo, blastocyst, or nonblastocyst. Sensitivity and specificity values of each embryologist were calculated.

**Implantation calculation**. The implantation outcomes of transferred embryos in the validation dataset containing 2086 videos were followed and calculated. After 5–6 weeks of transfer, the clinical pregnancy was determined by the presence of gestational sac with fetal heartbeat, and the implantation outcome of each blastocyst was calculated when the number of gestational sacs matches the number of transferred embryos. We compared the implantation rates between predicted usable blastocysts and unusable blastocysts by STEM$^+$ to evaluate the model's applicability.

**Reporting summary**. Further information on research design is available in the Nature Research Reporting Summary linked to this article.

## Data availability
The data underlying the findings of this study are available within the Supplementary Information and Supplementary Data 1. The videos for training and test sets used in this study were obtained with permission, stored in Mobile Hard disk belongs to our center, and are not publicly available.

## Code availability
The core codes used to train and evaluate the models are available from Github (https://github.com/bmelab2021/STEM).

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

## Acknowledgements

This work was supported by the National Key Technology R&D Program of China (2019YFC1005203, 2019YFC1005200, 2019YFC1005202, and 2018YFC1002103), Natural Science Foundation of Hubei Province (2017CFB800), Fundamental Research Funds for the Central Universities (HUST: 2020JYCXJJ015), and Health and Family Planning Commission of Hubei Province (WJ2019M127).

## Author contributions

K.L., S.X., and J.A. designed and supervised the study; Q.Liao and X.F. performed the data analysis and wrote the manuscript; Q.Z., H.H., and H.X. developed the algorithms and analyzed the data; B.T., J.L., and Q.Y. performed the modeling processes; N.G., Q. Liu, and B.H. labeled the videos. D.M. provided advices for experimental progress.

## Competing interests

The authors declare no competing interests.
