## [Peer Review File · Communications Biology]

Reviewers' comments:

Reviewer #1 (Remarks to the Author):

The manuscript by Liao et al. describes the author's investigations into predicting blastocyst formation on day 3. The author proposed an ensemble deep learning-based method and trained their models using 10,432 videos obtained from Time-lapse monitoring.

The manuscript is timely and is in line with the current trends in using deep-learning methods in IVF technology. Below I describe some points that deserve a better explanation or a different approach on the text. The criticisms below are provided to help the author enhance the quality of the manuscript:

(1) The manuscript requires proof reading and revision for some typos and improve the quality of writing. For example:

a. Line 31 in Abstract: "vide preparation algorithms" should be revised to "video preparation algorithms".

b. It's better to define acronym before using them. For example, in introduction (line 40), "day 5 or 6" should be first introduce as "day 5 or 6 (D5/6)" and then use it in line 46 as "D5/6".

c. Some acronyms such as ICSI should be defined first (line 108). I saw that it is defined later in method though.

d. The last paragraph of the introduction needs to be revised; it is hard to understand.

(2) I am wondering if the performance of ensemble algorithm (AUC = 0.82) is the result of validation set or the authors also used some of data as blind test. In case this result comes from validation, I recommend the authors to perform cross-validation across the dataset to show the robustness of trained model.

(3) The number of frames per video that were used for training is not clear to me. Different numbers were reported, and I am wondering if the final model benefits from 7 frames, 35 frames, or 600 frames? In case of using videos with 600 frames, how they were able to cope with memory limitation.

(4) I am also curious to know if the authors investigated performance of the same algorithm for Day 3 in comparison to Day 5 that usually used by embryologist for selecting the best embryo.

(5) I understand that predicting live birth is beyond the scope of this study. However, I am wondering if the authors have evaluated the association between the result of trained model and pregnancy likelihood.

(6) It seems the number of samples in blastocyst and non-blastocyst classes is not balance. I am wondering how the authors solved the unbalanced classes problem. Did they evaluate the effect of up-sampling technique, for instance, on their results?

(7) The manuscript needs more information about the training process. For instance, what optimizer they used for their model and what program language and framework they used for training their algorithm.

(8) If someone were to repeat the steps they took, they might not get very far with the level of detail that is provided in the methods section. An important issue to be rectified is that the codes is not provided for this study. I recommend the author to make a GitHub page for their algorithm. This could be very useful with a "Readme" file that indicates step by step of the training and evaluating the algorithm.

Reviewer #2 (Remarks to the Author):

In this analysis the authors have developed a model using deep learning networks to identify which human embryos from the 1-, 2-, 3-, 4- and 5-cell cleavage stage will form a blastocyst.

The Introduction is written with a distorted representation of the literature, suggesting an apparent urgent need to be able to predict if a cleavage stage embryo will form a blastocyst. This is simply not the case.

Being able to predict if an embryo will form a blastocyst is of little merit in human IVF as most all

embryos are typically cultured to day 5 as a matter of routine. Extended culture has been shown to be beneficial even for those patients with poor prognosis (Wilson et al., *Fertil Steril* 2002; 77: 693-6), and is associated with less pregnancy loss. So even in clinics which see an increase in no transfer rates when using blastocyst culture, the overall pregnancy rates go up (Marek et al., *Fertil Steril* 1999; 72: 1035-40). This indicates that an inability to develop to blastocyst in culture is diagnostic.

Furthermore, this model does not indicate the quality/score of the resultant blastocysts and hence has very little intrinsic value to clinical practise.

Although the algorithm appears to be well developed, it is 10 years out of date, and like the work of Wong et al., cited in reference #6, it has not been used to predict blastocyst quality or pregnancy outcome. Consequently, this work makes a limited contribution to the literature.

Reviewer #1:

The manuscript by Liao et al. describes the author's investigations into predicting blastocyst formation on day 3. The author proposed an ensemble deep learning-based method and trained their models using 10,432 videos obtained from Time-lapse monitoring.

The manuscript is timely and is in line with the current trends in using deep-learning methods in IVF technology. Below I describe some points that deserve a better explanation or a different approach on the text. The criticisms below are provided to help the author enhance the quality of the manuscript:

(1) The manuscript requires proof reading and revision for some typos and improve the quality of writing. For example:

a. Line 31 in Abstract: "vide preparation algorithms" should be revised to "video preparation algorithms".

b. It's better to define acronym before using them. For example, in introduction (line 40), "day 5 or 6" should be first introduce as "day 5 or 6 (D5/6)" and then use it in line 46 as "D5/6".

c. Some acronyms such as ICSI should be defined first (line 108). I saw that it is defined later in method though.

d. The last paragraph of the introduction needs to be revised; it is hard to understand.

Response: Thanks for your considerate suggestions. We have revised our mistakes as mentioned in a,b,c. And we simplified the last paragraph of introduction as follows

(lines 108-111): “Based on these insights, we aim to establish a model based on TLM videos and deep learning algorithms for accurately predicting blastocyst formation and blastocyst quality on D3. We believe this model will provide valuable information for clinical decisions about blastocyst culture and embryo selection.”

(2) I am wondering if the performance of ensemble algorithm (AUC = 0.82) is the result of validation set or the authors also used some of data as blind test. In case this result comes from validation, I recommend the authors to perform cross-validation across the dataset to show the robustness of trained model.

Response: Thanks for your suggestion. The performance of ensemble algorithm (AUC = 0.82) is the result of validation set (n=2086). We agree that use some of data as blind test or perform cross-validation will be more rigorous and robust. However, considering that the validation dataset was used only for test, and videos for test were in large quantities, we think the validation in these 2086 videos can show the model’s efficacy. In addition, it will take a relatively long time to implement the cross-validation because of the huge amount of video data, so we have not added it in our study this time. We can make further changes if the cross-validation is still needed.

(3) The number of frames per video that were used for training is not clear to me.

Different numbers were reported, and I am wondering if the final model benefits from 7 frames, 35 frames, or 600 frames? In case of using videos with 600 frames, how they were able to cope with memory limitation.

Response: Thank you very much for your comment. In the temporal stream model, we extract 600 frames ranging from PNF-100 to PNF+499 from each video for training. While in the spatial stream model, we chose 35 frames for training (there are five observation points in the first three days of embryo development and we extracted 7 frames at each point, $5 \times 7 = 35$). The final ensemble model was obtained by multiplying the output prediction value of the spatial and temporal stream model by their weight and then adding these values, so maybe we can say the final model benefits from both 600 frames (the temporal stream model) and 35 frames (the spatial stream model).

In temporal stream model where 600 frames were used, the frames were firstly input into the cell-counting model (with a suitable batch-size) and output as tag values composed of frame number and cell stage. Then, the LSTM network was employed to learn the tag values, not the frames, therefore we have no problems with memory limitation.

(4) I am also curious to know if the authors investigated performance of the same algorithm for Day 3 in comparison to Day 5 that usually used by embryologist for

selecting the best embryo.

Response: Thank you for your comment. Usually embryologists select good-quality embryos at D3 or D5/6. On D3, some embryos are selected for transfer into uterine immediately or vitrified. The remaining embryos are cultured to D5/6 to obtain blastocysts. If we were to investigate performance of our algorithm for Day 3 selection, the endpoint would be the clinical pregnancy or implantation rate of D3 transfer, which is not the same as our developed model (blastocyst as the endpoint). So we think the same algorithm can not be used to evaluate the efficacy of D3 selection directly. However, we believe our modeling methods can be applied to develop a model that help embryologists select high potential embryos for D3 transfer, and we're considering collecting videos and data for building this model.

(5) I understand that predicting live birth is beyond the scope of this study. However, I am wondering if the authors have evaluated the association between the result of trained model and pregnancy likelihood.

Response: Thank you for your valuable suggestion. We have revised our manuscript to add some clinical outcomes of the training model. In IVF clinics, embryos that cultured to D5/6 are scored by embryologists, only embryos with eligible expansion grade, inner cell mass (ICM) and trophectoderm (TE) can be used for transfer or vitrification, and they are defined as usable blastocysts. To improve the clinical application of our methods, we further developed a model to predict formation of

usable blastocyst. The newly developed model STEM+ reached 71.9% accuracy in predicting usability of embryos in validation videos (n=2086). As usable blastocysts can be transferred, we analyzed the implantation rate of these embryos. Only 209 blastocysts had known implantation outcomes (number of gestational sacs matched the number of transferred embryos). A total of 160 embryos were predicted as usable blastocysts by STEM+, of which 81 implanted and 79 failed. Another 49 embryos were predicted as unusable blastocysts by STEM+, and among them 21 implanted and 28 failed. The significantly higher implantation rate in predicted usable group (50.6% vs. 42.9%) indicated that STEM+ may help embryologists select embryos with higher development potential. The results of STEM+ and the associated implantation outcomes have been added in the last two paragraphs in Result as an independent portion “Usable blastocyst prediction and performance evaluation” (lines 248-271).

(6) It seems the number of samples in blastocyst and non-blastocyst classes is not balance. I am wondering how the authors solved the unbalanced classes problem. Did they evaluate the effect of up-sampling technique, for instance, on their results?

Response: Thanks for your suggestions. In the temporal stream model we adopted a weight adjustment strategy to balance the number of blastocyst and the nonblastocyst samples. In the spatial stream model we used the SMOTE algorithm to overcome the problem of unbalanced classes during the training process. And the results have a certain improvement after we used such technique.

(7) The manuscript needs more information about the training process. For instance, what optimizer they used for their model and what program language and framework they used for training their algorithm.

Response: Thanks for your considerable suggestions. For our model, Adam was used as optimizer, python was our program language, and we used Keras and Scikit-learn as the framework. These information has been provided in the “Video preparation” part in Methods (lines 439-440): “To develop our model, we used python as program language. The frameworks were Keras and Scikit-learn, and the optimizer was Adam”. We have to explain that the manuscript was drafted mainly by clinical doctors, so we may lose some important computing information about the training process. If more information is needed, we are glad to make further revisions.

(8) If someone were to repeat the steps they took, they might not get very far with the level of detail that is provided in the methods section. An important issue to be rectified is that the codes is not provided for this study. I recommend the author to make a GitHub page for their algorithm. This could be very useful with a “Readme” file that indicates step by step of the training and evaluating the algorithm.

Response: Thank you very much for your suggestion. We have organized the related codes and files for sharing. However, as we are applying for a patent on this model

now, we haven't provided these codes on GitHub. These codes will be publicly available as soon as we get the patent.

Reviewer #2:

In this analysis the authors have developed a model using deep learning networks to identify which human embryos from the 1-, 2-, 3-, 4- and 5-cell cleavage stage will form a blastocyst.

The Introduction is written with a distorted representation of the literature, suggesting an apparent urgent need to be able to predict if a cleavage stage embryo will form a blastocyst. This is simply not the case.

Being able to predict if an embryo will form a blastocyst is of little merit in human IVF as most all embryos are typically cultured to day 5 as a matter of routine. Extended culture has been shown to be beneficial even for those patients with poor prognosis (Wilson et al., *Fertil Steril* 2002; 77: 693-6), and is associated with less pregnancy loss. So even in clinics which see an increase in no transfer rates when using blastocyst culture, the overall pregnancy rates go up (Marek et al., *Fertil Steril* 1999; 72: 1035-40). This indicates that an inability to develop to blastocyst in culture is diagnostic.

Furthermore, this model does not indicate the quality/score of the resultant blastocysts and hence has very little intrinsic value to clinical practise.

Although the algorithm appears to be well developed, it is 10 years out of date, and like the work of Wong et al., cited in reference #6, it has not been used to predict blastocyst quality or pregnancy outcome. Consequently, this work makes a limited contribution to the literature.

Response: Thank you for your comments. To increase the clinical implication of our model, we further used this model to predict the formation of usable blastocyst, and the accuracy of STEM⁺ reached 71.9% in validation dataset. We compared the implantation rate of transferred embryos in validation dataset, the implantation rate of usable blastocyst predicted by STEM⁺ is much higher than unusable blastocyst predicted by STEM (50.6% vs. 42.9%), which indicated that our model can help embryologists select embryos with high implantation potential and improve the implantation rate in clinics.

For the description of predicting blastocyst formation in Introduction, We have made some changes according to the clinical practise in our country and the articles reporting blastocyst transfer vs. cleavage embryo transfer in PubMed.

“Most all embryos are typically cultured to day 5 as a matter of routine” is not the truth in our country (China), as embryos can be transferred either at cleavage stage (D3) or blastocyst stage (D5/6). For patients that good quality of cleavage-stage embryos are available, transfers are usually conducted at D3, and the remaining embryos are evaluated (mainly depends on patient’s age, number of embryos, embryo quality and so on) then cryopreserved on D3 or cultured to blastocysts for

cryopreservation. For patients with few good-quality cleavage embryos, there is a dilemma regarding when to transfer. Culturing these embryos to blastocysts allows for self-selection of embryos with good developmental potential and improves the pregnancy rates, but it may lead to cycle cancellation due to failure of embryo development to blastocyst stage. While cleavage-stage transfer can reduce the rate of transfer cancellation, and some embryos may develop better in vivo than in vitro, but implantation failures, biochemical pregnancies, and miscarriages of these poor-quality embryos are higher than blastocyst transfer. Therefore, an algorithm providing prediction of usable blastocyst formation can help embryologists choose patients or embryos suitable for extended culture, select embryos with high developmental potential, and make individualized clinical decisions.

In the literature, the controversy over cleavage-stage or blastocyst-stage transfer is not over yet. A Cochrane systematic review of 27 RCTs published in 2016 (Glujovsky et al., Cochrane database of systematic reviews 2016; 6(6): CD002118) reported that, although there is a benefit favouring blastocyst transfer in fresh cycles, it remains unclear whether the day of transfer impacts cumulative live birth and pregnancy rates. A population-based retrospective cohort study published in 2020 (Cameron et al., Human Reproduction 2020; 10:10) showed that there was not enough evidence to suggest a difference in the live birth rates following blastocyst- versus cleavage-stage embryo transfer in the first complete cycle of IVF. And we also find a clinical trial (Identifier:NCT03764865) (Neuhauser et al., 2020; 17(1)) is in progress investigating the clinical differences between cleavage- and blastocyst-stage transfer

in poorer prognosis patients. These latest articles indicate that blastocyst stage transfer is not a matter of routine in the literature, and our algorithm is not out of date.

Based on these thoughts, we revised Paragraph 1 in Introduction (lines 41-53) as follows: “Improving the efficacy of in vitro fertilization (IVF) has always been a focused issue, in which embryo selection and transfer are key procedures. Clinically, embryos are evaluated by embryologists and transferred at cleavage stage on Day 3 (D3) or blastocyst stage on Day 5 or 6 (D5/6) post-fertilization. Blastocyst transfer allows self-selection of embryos with higher developmental potential, thus can maximize the implantation rates and live birth rate in fresh cycles. However, it may lead to cycle cancellation due to failure of embryo development to blastocyst stage, causing mental stress of both patients and embryologists. Therefore, selecting embryos suitable for extended culture can help embryologists improve implantation rates, decrease transfer cancellation rate, and reduce mental stress.”

REVIEWERS' COMMENTS:

Reviewer #1 (Remarks to the Author):

Thanks for applying my comments and revise the manuscript. I hope you will be able to make your method publicly available in the future.

My suggestion is to provide the GitHub page with the codes and process, then you can add your trained model to the GitHub after the patent process is completed.

Reviewer #1 (Remarks to the Author):

Thanks for applying my comments and revise the manuscript. I hope you will be able to make your method publicly available in the future.

My suggestion is to provide the GitHub page with the codes and process, then you can add your trained model to the GitHub after the patent process is completed.

Response:

Thanks for your suggestion, we have provided the main codes and process on GitHub, the URL is <https://github.com/bmelab2021/STEM>.